# Exceptional Sorption of Heavy Metals from Natural Water by Halloysite Particles: A New Prospect of Highly Efficient Water Remediation

**DOI:** 10.3390/nano13071162

**Published:** 2023-03-24

**Authors:** Michał Stor, Kamil Czelej, Andrzej Krasiński, Leon Gradoń

**Affiliations:** Faculty of Chemical and Process Engineering, Warsaw University of Technology, Waryńskiego 1, 00-645 Warsaw, Poland

**Keywords:** halloysite, heavy metal sorption, lead, cadmium, sorbent activation, DFT calculations

## Abstract

Halloysite particles, with their unique multilayer nanostructure, are demonstrated here as highly efficient and readily available sorbent of heavy metals that can be easily scaled up and used in large-scale water remediation facilities. The various methods of raw material purification were applied, and their effects were verified using techniques such as BET isotherm (determination of specific surface area and size of pores), XRF analysis (composition), and SEM imaging (determination of morphology). A series of adsorption experiments for aqueous solutions of metal ions (i.e., lead, cadmium) were carried out to quantify the sorption capacity of halloysite particles for selected heavy metals. The ability of adequately activated halloysite to efficiently remove heavy metal ions from water solutions was confirmed. The value of the zeta potential of raw and purified halloysite particles in water was determined. This enables us to understand its importance for the sorption of positively charged ions (metal, organics) at various pH values. The adsorption process conducted in the pH range of 6.0–6.5 showed significant improvement compared to the acidic conditions (pH value 3.0–3.5) and resulted in a high sorption capacity of lead ions—above 24.3 mg/g for the sulphuric acid-treated sample. The atomic scale ab initio calculations revealed a significant difference in adsorption energy between the external siloxane surface and cross-sectional interlayer surface, resulting in pronounced adsorption anisotropy. A low energy barrier was calculated for the interlayer migration of heavy metals into the halloysite interior, facilitating access to the active sites in these regions, thus significantly increasing the sorption capacity and kinetics. DFT (density functional theory) calculations supporting this study allowed for predicting the sorption potential of pure halloysite structure towards heavy metals. To confront it with experimental results, it was crucial to determine proper purification conditions to obtain such a developed structure from the mineral ore. The results show a massive increase in the BET area and confirm a high sorption potential of modified halloysite towards heavy metals.

## 1. Introduction

Water is a key chemical compound on Earth, and its role for most living organisms is irreplaceable. Approximately 71% of the planet’s surface is covered by water, but only 3.5% of the volume of water is nonoceanic [1]. Due to global warming, the water cycle is intensified, which develops an imbalance in different Earth zones, making rainy regions more humid and dry areas even hotter [2]. On the scale of the globe, agriculture (including aquaculture and livestock breeding) is responsible for 69% of water demand, reaching 95% in some developing countries. Energy, power generation, and other industries are responsible for 19% of water usage, and the rest is used by municipalities [3]. A global environmental problem is water contamination by various undesirable compounds—APIs (Active Pharmaceuticals Ingredients), steroids, hormones, personal care products, surfactants, industrial and chemical additives, gasoline additives and pesticides and herbicides, heavy metal ions, and microplastics [4]. Minimizing pollution at the stage of water discharge from the factories and municipal plants is the first step to preventing the observed problem. The presence of contaminants causes serious health concerns. Most heavy metals such as mercury, lead, chromium, cadmium, and arsenic in drinking water affect human health by causing various dysfunction or damage of the organs, metabolism disorders, and other diseases, e.g., anemia, skin lesions, and vascular damage [5]. Exposure to high doses of mercury and lead induces health complications such as abdominal colic pain, bloody diarrhea, and kidney failure [6,7]. Chronic exposure to low doses of heavy metals might be diagnosed by its complications in the form of neuropsychiatric disorders [8], and some of them cause cancerogenic effects, misfunctioning DNA synthesis, and repair [9,10]. Based on global data collected over five decades (from 1972 to 2017), the concentration of heavy metals in natural waters has raised significantly, exceeding thresholds estimated by WHO (World Health Organization) [11]. The mean concentration of lead in the global river and lake water in the 1970s was at the level of 9.38 ± 4.60 µg L^−1^ and has increased 50 times in 90 years (257.62 ± 52.97 µg L^−1^) to achieve the top value in the collected data (Table 1). The latest value was estimated at 116.13 ± 25.48 µg L^−1^ [11]. A similar trend was also observed for cadmium and zinc. The highest concentration of copper was noticed in the 2000s and, for the research, the period resulted under WHO thresholds. The latest information about the concentration of mercury in surface water appears to exceed standards at least five times. The source of metal pollution was identified in the mining and manufacturing industries.

Until now, engineers have developed various methods for purifying water, which was crucial for urbanization or demanding industry zones. Different techniques like reverse osmosis [12], active sludge sorption [13], packed bed filtration [14], coagulation [15], biological treatment [16], or ozonation [17] are used in municipal water reclamation systems to deliver potable water to the households. Packed bed filtration is the technology that uses immobilized beds of sorbent. To intensify the process and reach higher sorption efficiency, a fluidized bed column with effective sorbent components should be considered. Proper material selection is critical for the removal of solid particles and the simultaneous adsorption of ions and organic compounds. There is a wide variety of sorbents available on the market to be used in this type of installation: active carbon [18], zeolites [19], clays [20], and other natural minerals. The activated carbon comprises various sorbent types (depending on the activation method and post-treatment applied), characterized by a high sorption capacity and well-developed surface area. It is, however, prone to colonization by microorganisms as it does not provide bacteriostatic properties [21]. Typical columns filled with activated carbon need to be frequently replaced or regenerated, which can also be done by including treatment with ozone to eliminate microorganisms that proliferate in the bed volume [22,23].

Halloysite is one of the clay minerals that exhibits properties that are typical for this type of natural material—good mechanical strength, relatively large pore volumes, high surface area, and chemical inertness [24,25,26]. The halloysite belongs to the kaolin-group minerals. The general formula can be expressed as Al_2_Si_2_O_5_(OH)_4_·2H_2_O in hydrated form. The crystal structure of halloysite is defined as a two-layer structure formed by a corner-sharing [SiO_4_] tetrahedral layer and an edge-sharing [AlO_6_] octahedral layer (as shown in Figure 1) [27]. Deposits of halloysite can be found in just a few parts of the world. Significant mines are in Australia, North America, Poland, and China. Depending on the origin of the mineral, various chemical compositions of the ore are reported—it may contain up to 12.8 mass% of iron oxides [28].

Unique for discussed material is a hollow nanotubular structure. The tubes have the spiral wound form and contain an interlayer zone, which can contribute greatly to the total specific surface area of the material, provided it is properly purified, which makes these microregions accessible. Due to its properties, halloysite finds application in fast-developing fields of nanotechnology like catalysis, drug treatment, or the production of other functional composites [29]. The role of the halloysite as an efficient sorbent was proven for a wide range of contaminations present in water, like pharmaceuticals [30], organic solvents [31], ammonium [32], and metal ions [33]. The usage of halloysite for heavy metal sorption was previously investigated [34] with raw material and composites [35]. The provided results exhibited a high potential for lead and cadmium adsorption on clay. The study was limited to only one purification method—sulphuric acid treatment—which succeeded in a sorption capacity 42.95 mg/g for lead ions. The unique structure form of halloysite makes it suitable for physical and chemical modifications, which allows for optimizing the material for a specific use. On the other hand, the inherent properties of the original material in properly prepared form (but without any secondary modification) make it an attractive and competitive alternative to other sorbents. Further applications in water purification are possible due to the low cost (compared to activated carbon), rich resources of this raw material [36,37], and potential photocatalytic properties [38].

This article describes research on structural characterization and analysis of the chemical composition of halloysite minerals before and after purification using various treatment conditions. In addition, the morphology of processed material was analyzed in Section 3.1. The adsorption efficiency of selected heavy metal ions was investigated and compared with results presented in the literature in Section 3.2. Finally, ab initio density functional theory was applied to determine the adsorption energy and the migration barrier of the selected heavy metals in the halloysite structure, which are crucial for sorption.

## 2. Materials and Methods

### 2.1. Materials

Raw halloysite used in this study was obtained from the Dunino mine (Legnica, Lower Silesia, Poland) [36,37]. The Dunino mine is an active, open pit type with deposits containing 10 million tons of naturally mixed raw minerals. A native material was sieved to isolate fractions 0.3–0.4 mm. Aqueous solutions containing heavy metal ions were purchased and diluted with deionized water: lead standard solution 1000 mg/L (No. 1.19776.0500) and cadmium standard solution 1000 mg/L (No. 1.19777.0500), which were both purchased from Merck KGaA (Darmstadt, Germany). Sulphuric acid solution 95% pure p.a., hydrochloric acid 35–38% pure p.a., and hydrogen peroxide 30% pure solution, all purchased from Pol-Aura (Zabrze, Poland), were used as halloysite purification reagents.

### 2.2. Halloysite Purification

The material purification process was conducted in five variants to estimate the optimal preparation for adsorption experiments. Solutions known for their etching properties (strong oxidizing acids and alkali) were chosen for the experiments to alter the halloysite microstructure. The primary goal of this modification was to improve the sorption process by increasing the specific surface area and changing morphology (i.e., contributions of micropores in the structure). Each batch was stirred on a magnetic stirrer for one hour in the following conditions: (1) 20% mas. hydrochloric acid solution at room temperature, (2) 20% mas. sulphuric acid solution at room temperature (3) 20% mas. hydrogen peroxide at room temperature, (4) 25% mas. sulphuric acid at a temperature elevated to 70 °C, (5) 50% mas. sulphuric acid at a temperature elevated to 100 °C. The glass vessels were thermally isolated to prevent the whole volume of the liquid (200 mL) from cooling down. Every solution was prepared with deionized water and kept in sealed containers to prevent loss of substance. The approximate amount of 20 g of raw halloysite per single batch was used. After the process, the material was rinsed with distilled water multiple times until a neutral pH value of water was obtained and was then dried for 4 h at 70 °C in the laboratory dryer. Details of treatment conditions are presented in Table 2.

### 2.3. Material Characterization Methods

The elemental composition of raw halloysite, along with purified material, was performed using an XRF (X-ray fluorescence spectrometer Fischerscope XRAY XDV-SDD, Helmut Fischer, Sindelfingen, Germany). The morphology of the halloysite samples was investigated using scanning electron microscopy (Hitachi SU-70) imaging with energy dispersive X-ray (SEM-EDX, Hitachi, Tokyo, Japan). The specific surface area was determined by fitting BET isotherm for the nitrogen adsorption–desorption process (3Flex, Micromeritics, Norcross, GA, USA). The zeta potential of halloysite particles was measured in water using Zetasizer NanoZS (Malvern Panalytical, Malvern, UK).

### 2.4. Adsorption Experiments

The adsorption properties of the material were evaluated in a batch system. Magnetic stirring of the sorbent suspension was applied, and all tests were conducted at room temperature. A precisely defined amount of sorbent (mass per volume of water contaminated with heavy metal ions) was prepared in 400 mL beakers. The initial concentration of contaminants in the water, the mass of sorbent particles per volume of water, and process time were determined based on the preliminary research results carried out in the past. The pH value of the water solutions was stabilized to be in the range of 6.0–6.5. The experiment was initiated when the halloysite particles were added to water. The sorbent particles were suspended by agitation; the processing time was 60 min. After this time, the particles were separated using a laboratory centrifuge, and the supernatant (clear water) was collected for analysis. Concentrations of heavy metal ions in liquid samples were analyzed using atomic absorption spectroscopy (AAS PinAAcle^®^ 900F, PerkinElmer, Waltham, MA, USA). Initial concentrations of metal ions solutions were selected to be 10.0 mg/L for lead and 2.0 mg/L for cadmium ions, respectively. The 60 min experiment time was assumed to be sufficient to reach a terminal concentration (sorbent saturation or depletion of the contaminant in the solution), which was confirmed experimentally for each ion type (based on preliminary studies, in which the transient concentrations between 5 min to 240 min were estimated, and no noticeable change of concentration was observed after 60 min). In the case of a lead solution, 0.2 g of sorbent was used, while the cadmium mass of sorbent material was 2.0 g per 200 mL of the solution (the difference results from various sorption capacities for selected ions which were used to maximize the accuracy of analysis).

### 2.5. Density Functional Theory Calculations

All ab initio calculations were performed using the density functional theory (DFT) method within the generalized Kohn–Sham scheme, as implemented in the VASP code [39]. The generalized gradient approximation (GGA) with the Perdew–Burke–Ernzerhof (PBE) functional was applied to treat the exchange and correlation energy [40]. The interaction of core and valence electrons was simulated with the projector augmented wave (PAW) potentials [41]. Valence wave functions were expanded as a linear combination of plane waves with an energy cutoff of 450 eV. The halloysite (7 Å) free surfaces were represented by supercells containing two [SiO_4_] tetrahedral plus two [AlO_6_] octahedral layers in the case of external siloxane termination and one [SiO_4_] tetrahedral plus one [AlO_6_] octahedral layer in case of cross-sectional interlayer termination, and they were separated by a vacuum space of 20 Å. Subsequently, the relaxed free surfaces were loaded with Pb, Hg, and Cd atoms and were subjected to relaxation with final convergence criteria of 1 × 10^−5^ eV and 0.03 eV/Å for total energy and force, respectively. A gamma-centered Monkhorst-Pack 4 × 4 × 2 sampling of the first Brillouin zone was applied to compute the total energy of the system. The minimum energy path (MEP) and the migration energy barrier simulations were performed with the climbing image nudged elastic band (CI-NEB) method.

## 3. Results and Discussions

### 3.1. Morphological and Chemical Characterizations

In Figure 2A–D, the SEM images of raw halloysite and the modified variant of the material are presented. The macroscopic surface differs depending on the purification method. Activated structures, compared to native material, demonstrate irregular, solid porous structures with smooth edges. The exception is the H3 sample, which is modified with a hydrogen peroxide solution. The structure was etched similarly to the other samples with fewer irregular, sharp edges, but most of the pores were closed. The SEM images of the H5 sample treated in a very aggressive chemical condition revealed the transformation of the surface to irregular and highly porous morphology. In Figure 2A–D the comparison between raw and activated halloysite is presented, for which a significant development of the internal porosity occurred. In addition, halloysite nanotubes (HNTs) can be identified in the structure of purified material. According to the literature, the dimensions of halloysite tubes are in the range of 0.02–30 µm length, and the outer diameter is from 0.05 to 0.2 µm [26]. Halloysite samples used in this research fall into the presented ranges. The tubular shape is noticeably mixed with various sheet shape particles. The presence of tubular structures indicates that purification with sulphuric acid does not affect the form of the mineral. However, it does not provide information about quantitative changes in the structures. Figure 2E presents incremental surface area for chosen (H0 and H5) samples, where a change in the pore width distribution is attributed to acidic treatment. In addition, the EDX analysis was conducted and presented in Table 3. Samples for SEM-EDX analysis did not contain large magnetic particles, and no additional preparation was needed. The analysis confirmed that fundamental elements of halloysite are oxygen, aluminum, and silica with traceable amounts of iron and titanium magnesium, which are considered to be natural impurities. As removing metallic contamination is essential in this study, the percentage ratio of impurities in purified material is an important indicator. To judge the removal efficiency of a given element properly, its relative concentration with regard to aluminum was calculated. The reduction of iron was revealed for each purification method, with the highest efficiency for the H4 sample and titanium for the H3 sample.

The EDX analysis proved the reduction of impurities from the clay for all purification methods considered in this study with various efficiency. The XRF analysis was conducted to further verify the chemical composition change.

Samples of powder clay were placed directly into the XRF apparatus with a standard setup for powdery materials. The time of the measurement was 40 s. For each sample five measurements were conducted, and the results were averaged. The X-ray fluorescence results presented in Table 4 confirmed the reduction of impurities registered in the EDX analysis. The composition of the raw halloysite is typical for clay materials. It should be emphasized that the XRF tests cover a relatively large area, and analysis is less focused on local differences in the impurities content, providing general information. A noticeable reduction of Fe content occurred in each purification method, with the best result for the H4 sample. Magnesium is missing in the XRF analysis due to the limitation of the technique.

An investigation of porous structures was conducted to evaluate the effect of the applied purification method on clay samples. Specifically, the BET method was used to analyze specific surface area (SSA) and porosity (Table 5). Two ranges of pore diameters can be considered, i.e., 0.5–2 nm (micropores) and 2–40 nm (mesopores). The specific surface area of raw material equals 32.3 m^2^/g. Modifications aimed at achieving the highest possible increment of SSA with regard to unmodified material to improve the sorption process. Samples processed in sulphuric acid at room temp. (H2), 70 °C (H4), and 100 °C (H5) resulted in SSA parameters that were higher than native material and achieved 67.4 m^2^/g (H2) and 74.5 m^2^/g (H4), respectively. The highest development of SSA was obtained for the H5 sample with a value raised to 162.6 m^2^/g and a total pore volume of 206%. The micropores volume in treated samples was raised (up to 0.026 cm^3^/g for H5), except for samples purified in hydrogen peroxide, where the contribution of micropores in total pore volume was lower than raw material and was stated to be less than 1.2% in total pore volume.

The chemical purification affected both the chemical composition and the structure of the halloysite. The reduction of the impurities, mostly iron, and the enhancement of surface parameters create better conditions for the sorption process. The purified material with the most favorable results is halloysite treated in a sulphuric acid solution with a temperature elevated to 100 °C due to the highest gained specific surface area (162.6 m^2^/g), the highest micropore volume (0.026 cm^3^/g), and nearly the lowest amount of iron in all variants of post-treated samples. The case of the H3 sample is worth additional analysis, as the effects are opposite than expected. A possible explanation of this result would be the specific impact of highly oxidizing hydrogen peroxide, which, in contact with hematite, can trigger the following chemical reaction, 2Fe_2_O_3_ + 6H_2_O_2_ → 4Fe(OH)_3_ + 3O_2_, and the formation of insoluble Fe(OH)_3_. The Fe(OH)_3_, in turn, is instantly deposited inside the pores and, due to the larger specific volume with regard to hematite, reduces the porosity of the halloysite. The results of the adsorption experiments described in Section 3.2 confirm the dependence of structure and chemical composition on adsorption properties.

### 3.2. Adsorption Experiments

To demonstrate the impact of the halloysite purification method on the sorption performance of heavy metals, the raw material and the samples H5 (for which the best structural parameters were achieved) were tested separately as sorbents for lead and cadmium from water solutions. The first set of adsorption experiments was carried out in acidic conditions that are typical for a nitrate-based solution (diluted standards), for which the pH range is 3.0–3.5. As presented in Figure 3, the sorption capacity of treated material was significantly lower than native sorbent. The results for low pH value indicated that the purification process had an adverse effect on the adsorption properties of the material at this pH value range.

The zeta potential for two selected halloysite samples, H0 and H5, was measured, which is a key parameter affecting the sorption. The particle suspensions were prepared using deionized water; the pH value was stabilized with 0.1 M water solutions of sulphuric acid and sodium hydroxide. Prior to the measurement in a standard 1 mL cuvette, larger particles were separated in high-speed centrifuge. The zeta potential of halloysite particles is crucial for the efficient sorption of heavy metal cations. First, the particles should be negatively charged to trigger the electrostatic attraction between the positively charged metal ions and the negatively charged particles. The higher the negative value of zeta potential, the stronger the electrostatic attraction and, essentially, the faster the adsorption of cations on the halloysite surface. As this parameter depends on the conductivity of the liquid and its pH value, zeta potential was mapped within the pH range of 2–8. As can be seen in Figure 3, the isoelectric point of the treated H5 sample (for pH = 3.4) is shifted towards higher pH values with regard to native material (pH = 2.6). Although the beneficial negative polarization of the halloysite surface covers a wide range of pH values for both samples, the magnitude of zeta potential for the H5 is significantly larger, reaching −29.9 mV at pH = 7.8, which is almost triple that for native material. This measurement alone explains the substantially lower sorption capacity of treated material than native sorbent, as the treated sample likely exhibits positive polarization in the acidic conditions, resulting in electrostatic repulsion (see Figure 3). By contrast, the native sample is still negatively polarized within the 3.0–3.5 pH range. 

The results lead to the conclusion that the higher pH of the solution containing metal cations creates better conditions for the adsorption process. Nevertheless, the optimum pH range for the sorption on halloysite in the literature was advised to be 5.0 ± 0.2 [42,43,44].

Adsorption tests for lead and cadmium ions in higher pH values were conducted, and the results are presented in Figure 4. Changes in the condition drastically influenced the efficiency of the process for the H5 sample, confirming a pronounced pH effect on the zeta potential and adsorption of cations. The sorption capacity of Pb (II) measured for the purified sample was above 24.3 mg/g, exceeding the results for native material by almost 25 times. It was probably even higher because, after sorption, the Pb concentration of remaining lead ions dropped below the detection limit of the applied analytical method (as commented below in Table 6). Compared to the reference data presented in Figure 4, the change in sorption capacity concerning the purification method is drastically higher. However, the sorption process was much less efficient for experiments conducted at pH = 3.0–3.5. This can be easily explained by the Zeta potential curve (Figure 3) at the pH in this range, as the surface potential was close to neutral (or even positive for acid-activated material). In such conditions, the interaction with positive metal ions was relatively weak to enhance the transport of metal, or a repulsive coulombic interaction could occur.

In Table 6, the estimated sorption capacities of Pb(II) and Cd(II) are compared with literature data from recent years. The results are comparable, although not all material properties and experimental conditions were the same in the cited studies for proper and justified comparison of these figures.

The sorption capacity can be considered a reference value that enables comparison of sorbent capabilities to store a certain amount of contaminant. However, one should remember that it refers to specific experimental conditions to reach a steady-state value (process time, temperature, bulk concentration etc.). A much more important indicator is the residual amount (concentration) of heavy metals in treated water. In Table 7, the concentration of Pb and Cd, which remain in the water after the sorption process, are listed and compared with the maximum acceptable concentrations of those contaminants established. The efficiency of lead sorption meets the limiting concentration for potable water, as established in WHO and USEPA standards [11].

### 3.3. Theoretical Insight into Sorption Properties of Halloysite

As demonstrated in the experimental investigations, the porosity control of the halloysite structure is a key factor in increasing its sorption efficiency and can be achieved by a properly selected purification method. However, the halloysite structure at the subnanometer scale should be analyzed to assess its maximum sorption potential toward heavy metal ions. For this reason, we used the ab initio DFT method and studied the interaction between the selected heavy metals and halloysite-free surfaces. First, we calculated the adsorption energy of lead, mercury, and cadmium on two different terminations of halloysite crystal, i.e., (1) the external siloxane surface, and (2) the cross-sectional interlayer surface (see Figure 5).

The adsorption energy can be expressed as the difference between the total energy of a relaxed system with an adsorbed metal ion on the surface and the total energy of a separated system (compared in Table 8):$$E_{ad}=E_{slab+X}-E_{slab}-E_{X}$$

Negative adsorption energy values revealed the affinity of the halloysite surface to the investigated heavy metals. A very important observation is the strong adsorption anisotropy, resulting from the large energy difference between the adsorption on the external siloxane and the cross-sectional interlayer terminations. Based on this observation, the adsorption process of the analyzed heavy metals is more intensive on the cross-sectional interlayer surface of halloysite nanotubes. Therefore, maximizing the exposure of cross-sectional interlayer termination in the halloysite particles is beneficial.

To fully describe the sorption potential of halloysite (7 Å) towards heavy metals, an interlayer migration of the adsorbed metal was simulated using the ab initio transition state theory and CI-NEB method. This method allows for sampling the potential energy surface (PES) and determining the migration energy barrier. The initial and final state geometries used in CI-NEB calculations are depicted in Figure 6. 

The interlayer migration barriers of Pb, Hg, and Cd atoms in the halloysite (7 Å), together with ionic radii, are presented in Table 8.

The values of interlayer migration barriers are very small; therefore, infiltration of the halloysite interior by heavy metal ions should be greatly facilitated. We can estimate the migration rate at *T* = 300 K by applying the Arrhenius law:$$\Gamma =Ae^{-\frac{E_{a}}{k_{B}T}}$$

Assuming $A\;\approx 10^{13}s^{-1}$ as a rough estimation of the characteristic atomic vibrational frequency, the migration rate of Pb is $\Gamma^{Pb}=1.34\times 10^{8}s^{-1}$ or 134 MHz. This demonstrates that interlayer migration in the halloysite (7 Å) is indeed a very fast process. 

## 4. Conclusions

In this work, the treatment of halloysite was applied with various etchants and conditions to gain the most effective changes in the structure, but without any other secondary modification of the halloysite structure (such as impregnation or doping with functional particles or chemicals). SEM imaging provided information about the microstructure with irregular, sharp edges, and multiple open pores, which is a noticeable change with regard to the smooth surface of the raw halloysite. The basic form of halloysite nanotubes (HNTs) was revealed after purification was performed in harsh acidic conditions, which implies that treatment does not affect the basic nanostructure. The zeta potential analysis of halloysite particles explained drastically different adsorption capacities in acidic and near-neutral solutions of heavy metals. The adsorption capacity for Pb(II) and Cd(II) in acidic conditions did not exceed 0.62 mg/g, and even the best purification method (which had a well-developed surface area) had no positive effect. The adsorption process conducted in the pH range of 6.0–6.5 showed significant improvement compared to the acidic conditions (pH value 3.0–3.5). It resulted in a high sorption capacity of lead ions (above 24.3 mg/g for the H5 sample). The ab initio DFT calculations resulted in very low interlayer migration barriers of the investigated heavy metal ions, providing a rational explanation for the exceptional sorption capacity of halloysite when its cross-sectional interlayer surface is well-exposed.

The complex structure and resulting properties of halloysite make it prone to functionalizing for various, more sophisticated applications. This is a prospective area of future research on optimizing the material for specific applications, including functionalization of the halloysite for processes such as catalysis or photocatalysis integrated with the sorption.

## Figures and Tables

**Figure 1 nanomaterials-13-01162-f001:**
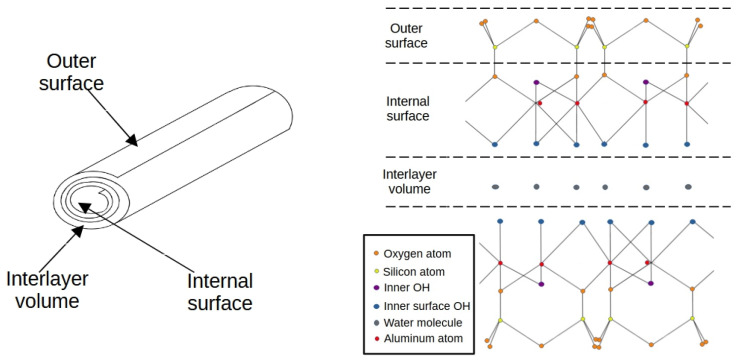
Halloysite structural scheme.

**Figure 2 nanomaterials-13-01162-f002:**
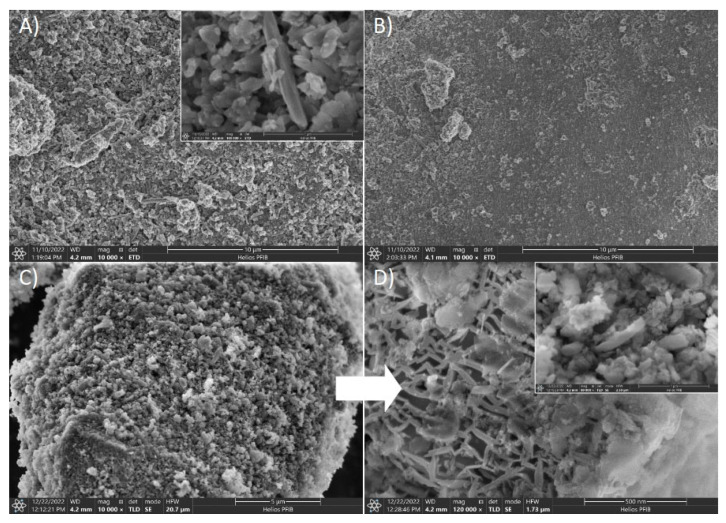
SEM Images of halloysite: (**A**) raw material (H0), (**B**) H_2_O_2_ treated sample (H3), (**C**) and (**D**) H_2_SO_4_ activated material (H5), (**E**) incremental surface area for raw (H0) and purified (H5) halloysite material.

**Figure 3 nanomaterials-13-01162-f003:**
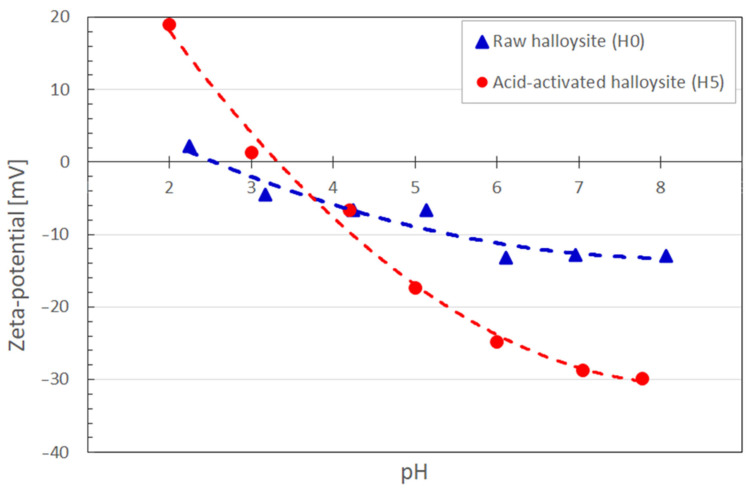
The influence of pH value on zeta potential for raw (H0) and acid-activated (H5) halloysite.

**Figure 4 nanomaterials-13-01162-f004:**
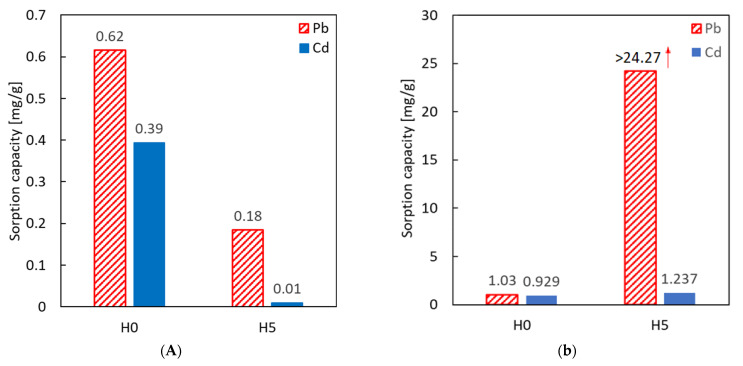
(**A**) Comparison of sorption capacity of Pb(II) and Cd(II) ions at pH 3.0–3.5 on the raw halloysite and on the material purified with 50% H_2_SO_4_ solution. (**B**) Sorption capacity of Pb(II) and Cd(II) in pH range stabilized to 6.0–6.5 value for raw halloysite (H0) and acid-treated halloysite (H5).

**Figure 5 nanomaterials-13-01162-f005:**
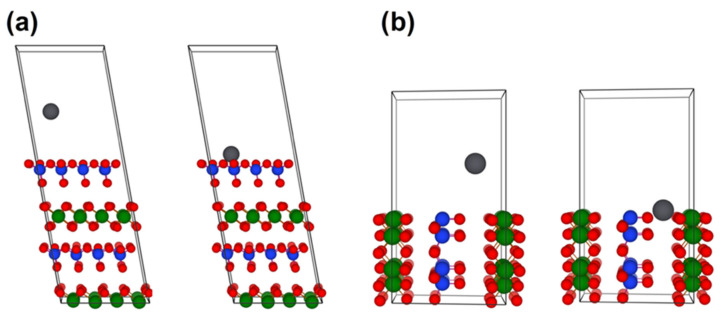
The slab supercell of halloysite (7 Å) structure that was used to simulate the adsorption energy of Pb, Hg, and Cd on: (**a**) external siloxane surface, (**b**) cross-sectional interlayer surface. Green, blue, and red spheres represent aluminum, silicon, and oxygen atoms, respectively.

**Figure 6 nanomaterials-13-01162-f006:**
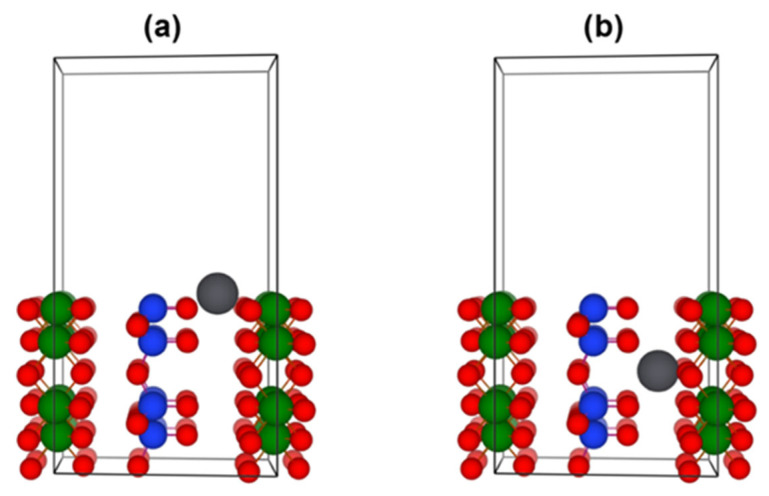
(**a**) Initial and (**b**) final state geometries used in the CI-NEB simulation of heavy metal ion interlayer migration in the halloysite (7 Å).

**Table 1 nanomaterials-13-01162-t001:** Total heavy metal concentrations (mg L^−1^) of the global river and lake water bodies from the 1970s to the 2010s, and concentrations of heavy metals in water as per World Health Organization (WHO) and United States Environmental Protection Agency (USEPA) standards [11].

Metals	1970s	1980s	1990s	2000s	2010s	Standards	
						WHO	USEPA
Cd	0.82 ± 018	0.74 ± 0.46	39.22 ± 10.29	21.60 ± 6.42	25.33 ± 7.17	3	5
Pb	9.38 ± 4.60	93.57 ± 90.45	257.62 ± 52.97	57.39 ± 18.91	116.13 ± 25.84	10	15
Hg	-	0.38	2480.00	3.91 ± 1.97	15.93 ± 9.96	1	2

**Table 2 nanomaterials-13-01162-t002:** Sample identification numbers and treatment conditions.

Sample Name	H0	H1	H2	H3	H4	H5
Etchant type	-	HCl	H_2_SO_4_	H_2_O_2_	H_2_SO_4_	H_2_SO_4_
Solution concentration [%mas.]	-	20	20	20	25	50
Temperature [°C]	-	22	22	22	70	100

**Table 3 nanomaterials-13-01162-t003:** The chemical composition of raw and modified halloysite samples obtained from SEM-EDX analysis. Values present a percentage of particular elements in the analyzed sample. Samples, treatment details, and names are described in Table 2.

	Sample Name					
	H0	H1	H2	H3	H4	H5
Element	[%]	[%]	[%]	[%]	[%]	[%]
O	57.52	60.92	62.12	58.44	63.19	62.78
Mg	0.70	0.31	0.305	0.105	0.135	0.34
Al	13.77	10.64	9.885	12.29	9.17	8.6
Si	16.91	16.39	14.50	17.16	20.27	24.41
K	0.33	0.23	0.105	0.05	0.065	0
Ca	0.30	0.17	0	0.25	0.045	0
Ti	0.75	0.60	0.35	0.235	0.78	1.63
Fe	4.78	3.57	4.075	2.705	1.36	1.79
Br	3.98	3.73	7.25	8.335	2.955	0
other	0.96	0.38	1.42	0.327	1.93	0.45

**Table 4 nanomaterials-13-01162-t004:** The chemical composition of halloysite raw and modified samples measured by the XRF method. Values present the percentage of particular elements in the analyzed sample. Samples, treatment details, and names are described in Table 2.

Element	H0	H1	H2	H3	H4	H5
Si	49.91	35.85	44.86	35.10	48.02	37.46
Al	29.67	56.95	38.59	49.09	40.22	52.84
Fe	17.71	5.60	14.57	12.68	2.99	4.12
Ti	1.93	0.80	1.31	1.15	1.27	2.38
Ni	0.16	0.02	0.03	0.05	0.06	0.01
Mn	0.08	0.01	0.14	0.10	0.00	0.03
Cr	0.04	0.03	0.07	0.06	0.02	-
Ca	0.41	0.09	0.20	0.56	0.09	0.11
S	-	0.47	-	1.12	7.25	2.77
other	0.09	0.17	0.24	0.09	0.08	0.27

**Table 5 nanomaterials-13-01162-t005:** Specific surface area (S_BET_), total pore volume (V_t_), mesopore volume (V_mes_), micropore volume (V_mic_), and average pore size (d) of raw and modified halloysite samples. Samples, treatment details, and names are described in Table 2.

Samples	BET Specific Surface Area [m^2^/g]	Total Pore Volume [cm^3^/g]	Micropore Surface Area [m^2^/g]	Mesopore Surface Area [m^2^/g]	Micropore Volume [cm^3^/g]	Mesopore Volume [cm^3^/g]	Average Pore Size [nm]
H0	32.2	0.186	33.35	45.17	0.005	0.114	15.73
H1	51.0	0.206	18.44	19.29	0.006	0.107	19.42
H2	67.4	0.221	18.65	31.43	0.007	0.133	14.16
H3	46.5	0.177	5.99	27.01	0.002	0.115	13.66
H4	74.5	0.205	35.56	21.61	0.013	0.108	15.26
H5	162.6	0.384	84.39	55.87	0.026	0.236	12.95

**Table 6 nanomaterials-13-01162-t006:** Sorption capacity calculated for the raw (H0) and sulphuric acid-treated (H5) sample of halloysite, compared with the reference values [34,35].

Halloysite Type	Metal	pH	Sorption Capacity	Ref.
Raw material (H0)	Pb(II)	3.0–3.5	0.62 [mg/g]	this work
Raw material (H0)		6.0–6.5	1.03 [mg/g]	this work
Raw material		5.0 ± 0.2	35.9 [mmol/kg] (7.44 [mg/g])	[34]
Acid-activated (H5)		3.0–3.5	0.18 [mg/g]	this work
Acid-activated (H5)		6.0–6.5	>24.3 [mg/g] *	this work
Acid-activated		5.0 ± 0.2	207.3 [mmol/kg] (42.95 [mg/g])	[34]
Calcinated		5.0 ± 0.2	207 [mmol/kg] (42.89 [mg/g])	[34]
Fe_3_O_4_ impregnated		5.0 ± 0.2	36.4 [mmol/kg] (7.54 [mg/g])	[35]
Raw material (H0)	Cd(II)	3.0–3.5	0.39 [mg/g]	this work
Raw material (H0)		6.0–6.5	0.93	this work
Raw material		5.0 ± 0.2	4 [mmol/kg] (0.83 [mg/g])	[34]
Acid-activated (H5)		3.0–3.5	0.11 [mg/g]	this work
Acid-activated (H5)		6.0–6.5	1.24	this work
Calcinated		5.0 ± 0.2	7.4 [mmol/kg] (1.53 [mg/g])	[34]
Fe_3_O_4_ impregnated		5.0 ± 0.2	10.8 [mmol/kg] (2.24 [mg/g])	[35]

* The concentration of Pb after the process was below the detection limit (the value of 0.01 (mg/L) was used to calculate the sorption capacity, which probably underestimates the real value).

**Table 7 nanomaterials-13-01162-t007:** Concentration of heavy metals after sorption. Concentration standards established by World Health Organization, United States Environmental Protection Agency, typical values for the Wastewater Treatment Plant discharge, results received in this study for raw halloysite (H0), and acid-treated (H5).

	Drinking Water		WWTP Discharge	Purified Water (Experimental)		
	WHO	EPA		Initial Conc.	H0	H5
Pb [mg/L]	0.010	0.015	0.1	10	4.83	<0.010
Cd [mg/L]	0.003	0.005	0.2	2.0	0.599	0.067

**Table 8 nanomaterials-13-01162-t008:** Adsorption energy of Pb, Hg, and Cd on the halloysite (7 Å) surface calculated with DFT and the interlayer migration barriers with ionic radii.

Metal	Adsorption Energy (eV)		Migration Barrier (eV)	Ionic Radii (pm)
	External Siloxane Surface	Cross-Sectional Interlayer Surface		
Pb	−1.09	−2.17	0.29	112
Hg	−0.49	−1.03	0.27	102
Cd	−0.56	−1.09	0.19	95

## Data Availability

Not applicable.

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
