# Peer review of "Exceptional Sorption of Heavy Metals from Natural Water by Halloysite Particles: A New Prospect of Highly Efficient Water Remediation"

_nanomaterials, 2023, doi:10.3390/nano13071162_

Round 1

Reviewer 1 Report

In the present work, the authors prepared five different halloysite based materials with different modification methods. And the obtained materials were used to remove heavy metal from water. BET and SEM were used to characterize the physical properties of the materials. And the adsorption capacity and adsorption energy were also investigated. My questions are as follows.

1.       Six samples including raw halloysite were used in the present work. However, the SEM results just show three samples, and BET results just give two. The other samples should be characterized by SEM and BET.

2.       The functional groups on the sample surface play important role in heavy metal removal process. FTIR will give some useful information.

3.       Why just H0 and H5 were chosen to have a heavy metal removal test?

4.       What kind of method did the author use to analyze the pore distribution?

Author Response

Review in attached file

Reviewer 2 Report

1.        The abstract needs to be made concise and key quantitative data should be included.

2.        The keyword “HNTs” should be removed to replace with two keywords “lead” and “cadmium”

3.        Section and sub-section numbers should be corrected for “Materials and methods” and “Results and discussion” as 2 and 3 respectively, as “Introduction” is already section 1.

4.        The information in paragraph given before the section “Materials” is not relevant to be put under the section “Materials and methods”.

5.        All the purchase details of chemicals/reagents and instruments/equipment/software/kits should be provided as state, city and country in the case of USA as well as city and country in the case of other countries. Also, for the second instance of same vendor/company’s mention, the authors can simply mention the company name, for instance, as Sigma-Aldrich and not very time Sigma-Aldrich (USA).

6.        The characterization methods involving XRF, SEM, EDX and zeta potential should be described in detail the sample preparation procedure and instrumental measurement parameters.

7.        Figure 4 and 6 should be combined and regardless of the color used, the bars pertaining to lead and cadmium should be differentiated by different pattern formatting.

8.        In Figure 5, the experimental data points should be differentiated with different legend markers for lead, and cadmium. For example, closed circles for lead and closed triangles for cadmium.

9.        Figure 2 and 3 should be combined as several parts of one figure.

10.     Absorption table and interlayer migration barrier table should be combined as one table.

11.     The abbreviations used in all the tables and figures should be explained in full form in the respective footers and captions. For example, H0 to H5, WHO, EPA, WWTP, & USEPA.

12.     The abbreviation of ‘minutes’ and ‘hours’ should be ‘min’ and ‘h’ respectively.

13.     The conclusion should be reduced for concise presentation.

Author Response

Review in attached file.

Reviewer 3 Report

The manuscript entitled « Exceptional sorption of heavy metals from natural water by halloysite particles: A new prospect of highly efficient water remediation” aimed on the purification of halloysite materials for heavy metal removal from water. The work is not original and the manuscript could be organized. The obtained results may interest the readers of the journal, and major revision are requested before possible publication:

1)      The abstract is too long and need to be shortened. Please cite carefully the novelty of this work in the abstract.

2)      The introduction should be revised to show the finding subject of this work.  Works on the utilization of mineral clay for the removal of heavy metals could be added and discussed in the introduction.

3)      From line 110 to 122 should be removed from the manuscript.

4)      The section (1.3. Halloysite purification) should be placed before the section (1.2. Material characterization methods).

5)      For the Halloysite purification; what is the weight of the raw materials that used for the different condition of purification? This section could be rewritten to show more details on the purification process.

6)      How to explain the change on the surface area with the condition of purification?

7)      If possible, the authors could study the kinetic of the heavy metal removal over the prepared halloysite.

8)      The conclusion is too long and should be shortened according the obtained interesting results in this work.

Author Response

Review in attached file.

Round 2

Reviewer 1 Report

It can be published now.

Author Response

Thank you very much for your effort, comments and useful remarks.

Reviewer 3 Report

1)      The full name of DEF should be added in the abstract.

2)      The keyword « HNTs » could be removed or supplemented in the abstract and then in the keywords list.

3)      Figure 3, the symbol ”[-]” associated to pH should be deleted.

Author Response

Thank you very much for your effort, comments and useful remarks.

We included all recommendations into final form of the manuscript.